# Lactobacilli Infection Case Reports in the Last Three Years and Safety Implications

**DOI:** 10.3390/nu14061178

**Published:** 2022-03-11

**Authors:** Franca Rossi, Carmela Amadoro, Maurizio Gasperi, Giampaolo Colavita

**Affiliations:** 1Diagnostica Specialistica, Sezione di Campobasso, Istituto Zooprofilattico Sperimentale dell’Abruzzo e del Molise “G. Caporale”, Via Garibaldi 155, 86100 Campobasso, Italy; 2Medicine and Health Science Department “V. Tiberio”, University of Molise, Via de Santis, 86100 Campobasso, Italy; carmela.amadoro@unimol.it (C.A.); gasperi@unimol.it (M.G.); colavita@unimol.it (G.C.)

**Keywords:** lactobacilli infections, update, case reports, virulence traits, safety implications

## Abstract

Lactobacilli constitute the dominant microbiota in many fermented foods and comprise widely used probiotics. However, these bacteria cause rare infections mostly in diabetic and immunocompromised subjects in presence of risk factors such as prosthetic hearth valves and dental procedures or caries. The scope of this survey was re-assessing the pathogenic potential of lactobacilli based on the infection case reports published in the last three years. In 2019, 2020, and 2021, total of 17, 15, and 16 cases, respectively, including endocarditis, bacteremia, and other infections, were reported. These annual numbers are higher than those observed previously. *Lacticaseibacillus rhamnosus* (13 cases), comprising strain GG (ATCC 53103) with established applications in healthcare, *L. paracasei* (7 cases), *Lactobacillus acidophilus* (5 cases), *L. jensenii* (5 cases), *Lactiplantibacillus plantarum* (3 cases), *L. paraplantarum*, *L. delbrueckii* subsp. *delbrueckii*, *L. gasseri*, *L. paragasseri*, *Limosilactobacillus fermentum*, and *L. reuteri* (1 case each) were involved. Virulence characterization of two strains that caused infections, a derivative of *L. rhamnosus* GG and *L. paracasei* LP10266, indicated that increased biofilm-forming capacity favors pathogenicity and it is determined by variable genetic traits. This survey highlights that the strains of lactobacilli that cause infections are little characterized genetically. Instead, to avoid that these bacteria become a hazard, genetic stability should be periodically re-evaluated by whole genome sequencing (WGS) to ensure that only non-pathogenic variants are administered to vulnerable individuals.

## 1. Introduction

Lactobacilli, i.e., bacteria belonging to species included in the genus “*Lactobacillus*” before it was divided in 25 new genera in 2020 [1], are Gram-positive anaerobic aerotolerant asporigenous bacteria with a rod cell shape. These are responsible for fermentative processes in foods where they convert carbohydrates to organic acids, ethanol, and CO_2_, with lactic acid as the sole or one of the main metabolic products. These bacteria are intimately connected with human wellbeing, since they are able to colonize different body sites such as the oropharynx, gastrointestinal tract (GIT), and female urogenital tract where they contrast pathogenic microorganisms. Lactobacilli are ingested through the consumption of traditional fermented products. Representatives of these food products are manufactured in all parts of the world, where they have been a component of human diet throughout millennia. In fermented foods lactobacilli release organic acids and other substances with antimicrobial properties that allow to obtain microbiologically safe, nutrient-rich fermented matrices of animal and plant origin. Lactobacilli enhance the nutritional value of these foods by increasing their digestibility, and act as a living component that helps preventing and treating many health disorders. In some countries, fermented foods were usually prepared according to traditional methods without addition of well-characterized bacterial cultures. These foods are perceived as safe by consumers and are commonly considered to contain living microorganisms with health-promoting effects [2].

Foods can benefit of the definition “probiotic” when they contain bacterial strains that exert beneficial effects on the host when administered in adequate amounts [3] and their beneficial effects are proven by clinical studies. This is also true for fermented foods. These can be defined “probiotic” if their beneficial effects are proven and depend on the presence of microbial strains with a probiotic action [4]. Single strains of lactobacilli have well-recognized effectiveness in the prevention and treatment of diseases of infectious or other nature and hold the generally recognized as safe (GRAS) status conferred by the Food and Drug Administration (FDA) in the USA [5]. In Europe, all species belonging to the old genus *Lactobacillus* reclassified in 2020 and intended for use in food products hold the qualified presumption of safety (QPS) status recognized by the European Food Safety Authority [6]. The market of probiotic lactobacilli has been growing constantly and has perspectives of further increases in the years to come [7], since these bacteria are used for the production of probiotic foods, food supplements, and pharmaceutical preparations designated as “live biotherapeutic products” (LBPs) that are intended for treatment of specific medical conditions. The latter may include one or more bacterial strains, must be stable in composition and efficacy, and must be safe on the basis of standardized evaluation protocols [8].

The assumption of probiotics is well accepted by consumers who are increasingly aware of their beneficial effects. However, current regulations still do not ensure that all the probiotics commercially available have a proven efficacy [9].

On the other hand, lactobacilli have also the potential to cause infections, that are considered to be rare on the basis of the number of case reports published until 2019. Infections caused by lactobacilli, have been mostly reported in immunocompromised and diabetic patients, but sometimes also in subjects without underlying conditions or risk factors [10].

This new survey of case reports, limited to the last three years, was carried out to collect updated knowledge on the involvement of lactobacilli in infections. The aim was to obtain indications on the current trend and on how to reduce the risk of disease that these bacteria cause when ingested with fermented food, probiotic food, food supplements, and LBPs. This aspect deserves attention especially because reported episodes of infections occurred primarily in patients with medical conditions, such as preterm birth, diabetes, and immunodeficiency in which probiotic lactobacilli constitute a valuable aid [11,12,13].

## 2. Cases of Lactobacilli Infections Reported in Years 2019, 2020 and 2021

In the period 2019–2021, 48 case reports of infections caused by lactobacilli, with an almost uniform annual distribution, were published. These were retrieved from scientific literature databases by using the search terms “*Lactobacillus*”, or any of the names of the genera newly classified in 2020 and comprising probiotics [1], followed by the terms “infection” or “bacteremia”, or “endocarditis”, or “abscess”. It must be underlined that none of the case reports or retrospective studies retrieved used the new nomenclature for lactobacilli. Among the case reports found, endocarditis and bacteremia predominated, according to previous records [10]. Table 1 shows the number of cases of endocarditis and bacteremia reported per year, together with the range of patients’ age, risk factors, underlying medical conditions, and identity of the etiological agents. The references are numbered according to the chronological order of appearance in journals or conference proceedings. Case reports published online in 2021 but appearing on journals in early 2022 are included as 2021 entries, as also shown in Appendix A.

In Table 2 the types of infection different from endocarditis and bacteremia attributed to lactobacilli in 2019, 2020, and 2021 are listed. Age of the patient, risk factors, underlying conditions, and species identity of the infectious agent are shown. Case reports published online in 2021, but appearing in journals in early 2022, are included as 2021 entries, as also in Appendix A.

The description of each case is detailed in Appendix A, with indication of the method of identification of the causative agent. In four cases regarding patients with severe underlying conditions the outcome was fatal, though the infection had been resolved by antibiotic treatment [29,36,38,44]. This accounted for a mortality rate of 8.3%.

For eighteen reports, the species involved was specified but the method of identification was not declared. For other nine reports, identification to the species level was not carried out and the agent of infection was designated as “*Lactobacillus* spp.”. One study reported just Gram staining as the method of identification [54]. In the case of interstitial pneumonia published in 2020 (Table 2) cultures were not carried out, but it was assumed that the *Lactiplantibacillus paraplantarum* strain present in the probiotic yogurt consumed by the patient was the cause, since ground-glass nodules (GGNs) formed in the lungs disappeared after discontinuing the assumption of the probiotic and vildagliptin [50].

Lactobacilli were isolated from blood [15,16,17,18,20,21,22,24,25,26,27,28,29,30,31,32,34,35,36,37,38,39,40,41,42,43,44,45,46,47,49,52,55,56,57], diseased valve tissue [23], abscess fluids [48,51,52,54,57,58], cerebrospinal fluid (CSF) [46], marrow [30], arthrocentesis aspiration fluid [53] in anaerobiosis and, in some cases, also in aerobiosis [15,34]. Use of automated microbial detection system, i.e., BACT/ALERT 3D or the BACTEC systems (BioMérieux) with incubation at 35 °C were mentioned in some reports [23,31,34]. The case reports that described the identification procedures mostly referred to standardized instrumental techniques widely adopted in clinical microbiology laboratories. These were matrix-assisted laser desorption/ionization time-of-flight (MALDI-TOF) mass spectrometry (Appendix A), that relies on the interpretation of spectra by reference to databases, or colorimetric methods, e.g., Vitek 2 automated identification system, with ANC ID card, (BioMérieux) [43,46], or both (Appendix A). In one instance, the RapID CB Plus system (Thermo Scientific) was used [39]. Sequencing of the 16S rRNA gene was carried out in six instances and whole genome sequencing in two (Appendix A). In one case there was discordance in the identification with MALDI TOF mass spectrometry and 16S rRNA gene sequencing between the strictly related species *L. gasseri* and *L. paragasseri* [58].

De Freitas et al. [49] stated that their facility was not equipped for identification of the level of species.

Antibiotic treatment of infections was always successful, though in some cases it was initiated empirically and changed later based on antibiotic susceptibility testing. Susceptibility to β-lactams, including amoxicillin, amoxicillin/clavulanic acid, ampicillin, ampicillin-sulbactam, benzylpenicillin, penicillin G, piperacillin/tazobactam, and meropenem IV was reported in most cases. These antibiotics were used singly or in association with gentamicin, clindamycin, and clarithromycin [16,17,20,21,22,26,27,28,29,30,31,32,35,36,37,38,39,40,43,44,45,46,48,49,51,53,54,55,56,57,58].

## 3. Aspects of the Lactobacilli Infections Cases

### 3.1. Bacteremia

In two cases bacteremia was associated to localized infections, namely, liver abscess [37] and urinary tract infection (UTI) [39]. In the case reported by Sendil et al. [41] bacteremia evolved to septic shock.

The origin of the infectious agent was investigated by Chiang et al., 2021 [45], who observed one case of bacteremia in a preterm girl and reviewed cases of neonates with *L. rhamnosus* GG bacteremia reported until November 2019. Beyond the new case, other eight reports were found, all occurring in 2004, in infants of less than three months of age. They could state that in at least 55.6% of the infants the infection originated from contamination of the central venous catheter (CVC) and in three cases the tip of the catheter grew *L. rhamnosus* GG. For the case reported by Celis Castaňeda et al. [36], it was specified that the *L. reuteri* probiotic was administered to the baby according to the institutional protocol. Moreover, in this case a CVC was used and mechanical ventilation was applied.

Haziri et al. [42] hypothesized that the origin of the infectious agent was a yogurt made at home by the patient, though isolates from yogurt were not examined.

### 3.2. Endocarditis

Most of the cases of endocarditis had a deceitful and slow onset, with or without fever, with symptoms varying among patients and including weight loss, fatigue, dyspnea, cough, chills, sweats, abdominal pain with nausea and vomiting, lumbar pain, syncope, with valvular vegetations becoming detectable after many days or even months from the beginning of symptoms [14,15,16,17,18,20,22,23,27,29,30,31,32,34]. In many cases replacement of the aortic valve [15,16,17,18,20,21,26,28,30,35], in one case replacement of the mitral valve [25], and in another case replacement of both aortic and mitral valves [23] were necessary.

One case occurring in a healthy 50-year-old male, who had as a possible risk factors, use of a probiotic supplement and a gingival laceration 3 months prior to the clinical manifestations of endocarditis, presented both a lesion of the native mitral valve and a perforation of the native aortic valve. The presence of splenic infarction possibly caused by septic emboli was also reported [23]. In the case reported by Campbell et al. [25] consumption of probiotic yogurt was the sole risk factor and native mitral valve was affected. Argotsinger et al. [33] reported native aortic valve infection in absence of risk factors and medical history.

Complications observed in cases of endocarditis comprised:Multilevel discitis [18,29];Occlusion of the superior mesenteric artery for a thrombus, requiring urgent surgical laparotomy, associated to emboli in the brain cortex [34];Splenic infarct, possibly caused by septic emboli [18,20,23];Multiple embolic strokes with acute-to-subacute infarct in the brain parietal lobe and a rapidly progressive glomerulonephritis (RPGN) [24];Splenic abscess [20];Glomerulonephritis and thrombotic microangiopathy [19];Microabscesses in the psoas [29];Embolic stroke and presumed lumbar vertebral osteomyelitis [33];Recurrent transient ischemic attacks possibly related to a central embolic source causing left side lower extremity weakness and expressive aphasia [15];Peroneal mycotic pseudoaneurysm supposed to be caused by seeding of blood-borne bacteria to the vasa vasorum of the artery wall or septic emboli from the infected heart valve [35];Acute respiratory failure and septic shock [17];Infarction secondary to septic embolism and ruptured mycotic brain aneurysm suspected to be caused by septic embolism from the aortic valve vegetation with right-sided hemiplegia and aphasia [14];Ankle arthritis with cutaneous eruption [22].

One case of septicemia following endocarditis attributed to *Lactobacillus jensenii* was consequent to asymptomatic urolithiasis with bilateral ureter obstruction [26].

### 3.3. Other Infections Caused by Lactobacilli and Complications

Case reports of localized infection caused by lactobacilli in the last three years are summarized in Table 2. Many presented complications such as the spreading of the infection to other body areas.

A 63-year-old patient affected by lung cancer, who developed meningoencephalitis caused by *L. plantarum*, also manifested atrial fibrillation, left atrium enlargement, and trace of mitral regurgitation, though with no evidence of active endocarditis [46].

Ranchal et al. [55] reported of a large abscess of the prostate caused by *L. jensenii* extending to the seminal vesicles and pelvic muscles and with bladder fistulization. The case was complicated by bacteremia, pulmonary septic emboli, and probable right-sided endocarditis.

In a 49-year-old patient, UTI caused by *L. delbrueckii* [48] was complicated by pyelitis with purulent discharge because of resistance of the causative strain to quinolones, that had been initially used to treat the infection. Recovery was obtained by administering cefotaxime and amoxicillin.

A case of perinephric abscess in a 26 years-old lady due to *L. jensenii* and *Prevotella bivia* was complicated by *L. jensenii* bacteremia [52].

A pyelonephritis case caused by unidentified lactobacilli, described by De Freitas et al. [49] was complicated by a perinephric abscess, never reported before to be caused by lactobacilli, and bacteremia.

Pancreatic necrosis attributed to *L. paracasei*, as reported by Miwa et al. [57], was associated with retroperitoneal abscess and bacteremia.

A case report not included in previous reviews but worth of being mentioned for its uniqueness and severity was published in 2018 [59] and regards fasciitis caused by *L. acidophilus* in a 59-year-old diabetic woman. The infection caused an abdominopelvic wound with necrotic tissue along the fascial planes. Repeated operative procedures were necessary to eliminate necrotic tissue that continued to form. Authors stated that cultures of the necrotic tissue revealed the presence of *L. acidophilus* but the identification method was not specified. Antibiotic treatment with doxycycline and ceftazidime allowed wound healing in thirty days.

## 4. Investigations on the Frequency of Lactobacilli Infections

Three recent studies analyzed retrospectively the occurrence of infections caused by lactobacilli in two hospitals [60,61] and the intensive care unit (ICU) of a children’s hospital [62] and found that numerous cases were recorded in four years, one year, and five years, respectively. Albarillo et al. [60] reported that a total of 47 patients had *L. rhamnosus* or *L. rhamnosus/casei* growth from different types of specimens, i.e., blood, abdominal fluid, abscess, pleural fluid, bronchial fluid, urine, and sputum. Since 2015, MALDI TOF spectrometry was used for isolate identification and in 12 cases the species *L. rhamnosus* and *paracasei* could not be distinguished. The average age of patients was 63 years with almost the same number of males and females. All the cases had polymicrobial infections and nine patients died due to underlying conditions rather than the *Lacticaseibacillus* infection. The authors concluded that these lactobacilli have low pathogenic potential.

Yelin et al. [62] analyzed records in a period of five years finding a significantly higher frequency (1.1%, 6 of 522 patients) of bacteremia caused by lactobacilli in patients who received the probiotic *L. rhamnosus* GG compared to those who did not (2 out of 21,652). Six isolates were obtained from patients with bacteremia in the group receiving the probiotic and all were identified as *L. rhamnosus* by MALDI-TOF mass spectrometry, while two isolates from the bacteremia cases of patients not receiving the probiotic were identified as other lactobacilli. Among non-ICU patients ten out of 93,000 who did not receive a probiotic had bacteremia caused by lactobacilli and the isolates from four of them were identified as *L. rhamnosus*, thus indicating that lactobacilli can cause bacteremia also in absence of probiotic supplementation, but at a much lower frequency. The four *L. rhamnosus* isolates from the non-ICU patients showed higher similarity of the genome sequence with other *L. rhamnosus* strains than with *L. rhamnosus* GG.

Nwanyanwu et al. [61] reported ten cases of bacteremia by lactobacilli in patients with no declared use of lactobacilli probiotics. These patients required long hospitalization (38.5 ± 27.6 days). The suspected sources of infection were the gastrointestinal tract in five cases, infective endocarditis in one case, and genitourinary tract in another case. The source of infection in three cases could not be determined. Four patients had a co-infection with *Candida* spp. and four with enteric bacteria. Four patients died.

In a study that exploited metagenomic analysis by DNA shotgun sequencing on excised valve tissue to identify the infectious agents responsible for blood culture-negative endocarditis (BCNE), the species *Limosilactobacillus fermentum* was detected in one among eleven valves from ten patients and was considered the causative agent, according to the criteria established in the study [63].

Since oral hygiene can have an impact on the occurrence of lactobacilli infections, in this survey we also considered studies on this aspect. In a study aimed at analyzing the distribution of beta-lactamase resistance genes in patients with periodontitis, it was found that lactobacilli, identified by 16S rRNA gene sequencing, were involved in 26.3% of 129 cases of periodontitis. *Limosilactobacillus fermentum* was most often isolated, followed by *Ligilactobacillus salivarius*, *L. paracasei*, and *L. rhamnosus* [64].

Finally, in a study that aimed to define the virulence of enterococci isolated from carious dentine, Ceccon Chianca et al. [65] found that all the isolates initially identified as *Enterococcus faecalis* by PCR were correctly identified as *Lactobacillus* spp. by MALDI-TOF. These isolates produced biofilm in the presence of saliva, were acidogenic, a trait involved in enamel demineralization, and could be therefore involved in caries formation.

## 5. Virulence Characters and Proposed Pathogenesis

Information on the physiological traits that can influence virulence in lactobacilli derived from some of the case reports surveyed in this article. Chiang et al., 2020 [45], who analyzed five isolates from a preterm girl with bacteremia and the administered probiotic, found that *L. rhamnosus* GG (ATCC 53103) and the administered probiotic strain, as well as isolates from blood and catheter tip, formed biofilms in all the growth conditions tested. Only a stool isolate did not form biofilm. Moreover, they observed that glucose enhanced biofilm formation. Thus, an infection dynamic was hypothesized that implies the possible bacterial translocation of *L. rhamnosus* GG across the immature gut epithelium of the preterm infant and enhancement of biofilm formation for the presence parenteral administration of glucose through the peripherally inserted central catheter (PICC). Four isolates from the patient (two from blood, one from the catheter tip and one from stool) and one isolate from the probiotic preparation shared five identical single nucleotide variations (SNVs) from the *L. rhamnosus* GG (ATCC 53103) genome available in the NCBI database. The isolate from the probiotic product exhibited an additional SNV. This finding demonstrated that short-term evolution of a probiotic can occur. One SNV common to the isolates was a nonsynonymous mutation T924G in the gene encoding a CamS family sex pheromone protein, with amino acid substitution H308Q. A different amino acid substitution in the same protein, H294Q, was reported by Yelin et al. [62] both in blood isolates and in isolates from the probiotic that was administered to the patients. Investigations on the functional role of this protein could help elucidate if it influences biofilm formation.

Yelin et al. [62] reported five additional mutations in isolates from blood, supporting rapid in vivo evolution of the probiotic. One of these mutations, implying the amino acid substitution H487D in the *rpo*B gene encoding RNA polymerase, conferred rifampin resistance and occurred in a patient who had received for three months rifaximin together with *L. rhamnosus* GG.

The role of the biofilm formation capacity in virulence was confirmed by Tang et al. [30]. They observed that *L. paracasei* LP10266, isolated from blood and marrow samples of a patient with endocarditis, did not induce platelet aggregation and induced complement activation. However, this strain displayed a strong biofilm formation ability and adherence to human vascular endothelial cells. This strain has two *spa*CBA pilus gene clusters and a novel exopolysaccharides (EPS) cluster. Relating to biofilm formation capacity, it was demonstrated that in *L. rhamnosus* GG the SpaCBA pilus has a key role in it [66], but its coding region can be lost, i.e., during yogurt production, in a minority of derivatives [67].

Zafar et al. [68] investigated the possible involvement of membrane transporters in the virulence of lactobacilli comparing six species supposed to have only a probiotic action (*L. brevis*, *L. delbrueckii* subsp. *bulgaricus*, *L. crispatus*, *L. gasseri*, *L. reuteri*, and *L. ruminis*) and four species involved in infection cases (*L. acidophilus*, *L. paracasei*, *L. plantarum*, and *L. rhamnosus*). They found that the latter species have a higher number of sugar, amino acid, peptide transporters, and drug exporters. Although also the species with so far unrecognized pathogenic potential contain pore-forming toxins and drug exporters similar to those of the probiotic and pathogenic species. In particular, *L. paracasei*, *L. plantarum*, and *L. rhamnosus* have a much higher number of drug exporters, amino acid transporters, sugar transporters, and unknown transporters.

Regarding patient susceptibility to lactobacilli infections, possible explanations were given by some of the case report authors. One is that diabetes mellitus, one of the main predisposing conditions, is associated with increased vascular permeability and non-occlusive microangiopathy consequent to the glycosylation of basement membranes. In addition, metabolic alterations in diabetes contribute to endothelial cell damage that could be a possible route of bacterial translocation to blood and from blood to other body sites [37]. Therefore, the combination of increasing probiotic use [7] and the increasing trend of diabetes incidence [69] could explain the rise of lactobacilli infections. In six cases reviewed here [20,28,37,39,57,59] it was stated that diabetes mellitus was poorly controlled by the patient, thus indicating that a better management of this condition could reduce the probability of lactobacilli infections. For what concerns patients with immunosuppression as a population at risk of lactobacilli infections, it must be considered that also this category is on the rise for the increase in use of immunosuppressive drugs to treat cancer, organ transplants, respiratory syndromes such as asthma and chronic obstructive pulmonary disease, dialysis, autoimmune diseases, inflammatory skin conditions, Crohn’s disease [70,71,72]. In an investigation published in 2018 [70] it was found that almost 1 in 5 persons in New York and 1 in 6 persons in Sydney, and even more for the 60–64-year age group, could be considered immunodepressed. Immunosuppressive drugs mostly used include antineoplastics, glucocorticoids, and noncorticosteroid immunosuppressants [71].

Chukwurah et al. [24] hypothesized that chronic use of ibuprofen could result in erosion of the lining of the GI tract, increasing the likelihood of the entry of lactobacilli originating from dental caries into the blood stream. Bacterial translocation from gut was deemed as possible also in other case reports [29,38,57].

According to Yelin et al. [62] the ICU pediatric patients who received *L. rhamnosus* GG most probably developed infection after contamination of the CVC, either directly with the probiotic strain or with stool containing the probiotic strain. However, translocation of the probiotic across the bowel wall was not excluded. No risk factors that could explain the onset of bacteremia only in some of the subjects could be identified.

## 6. Discussion

Insidious onset, severe symptomatology, long hospitalization, were common characteristics of lactobacilli infections reviewed here.

However, given the difficulty in defining a common disease progression and common clinical signs, taking into account routinely the eventuality of infection by lactobacilli is fundamental to allow a quicker identification of the etiological agent and start without delay an appropriate antibiotic treatment, thus enabling the mitigation or the prevention of severe consequences.

From the reports in this survey, it was deduced that the bacterial strains causing the cases did not present acquired antimicrobial resistance (AMR) traits, at least toward the antibiotics used to treat patients. Indeed, the susceptibility profile of the isolates was in line with what was reported previously for lactobacilli, i.e., intrinsic high level resistance to vancomycin and susceptibility to β-lactams. Susceptibility to quinupristin/dalfopristin, chloramphenicol and linezolid was also reported [73,74,75]. Unfortunately, in many of the cases considered here patients were started on vancomycin and cephalosporins and only later switched to β-lactams, thus probably contributing to disease worsening [20,29,32,37,38,39,40,41,45,46,47,49,52,53,56]. From the case reports, resistance of lactobacilli to cephalosporins appeared to be a common trait to be further investigated.

The mortality rate of 8.3% for the cases considered in this survey was similar to that reported by Campagne et al. [22] for endocarditis, that was as high as 10% until 2018. They also found that cases of endocarditis caused by lactobacilli clearly showed an increase of reports per year since 1992. Increased annual incidence was particularly high since 2016, with fourteen cases in years 2016–2018 against 38 cases published in the previous 23 years, almost triplicating the number of reported cases per year. The present survey shows that the number of endocarditis reports per year further increased after 2018, reaching 23 case reports in three years with a peak of ten in 2020. Explanations can be the increased consumption of probiotics [7] and a higher percentage of the population with predisposing conditions. A question to answer is if these bacteria can still be considered to cause “rare” infections, innocuous for the general population, and what is the definition of “general population” [6]. Indeed, risk factors such as prosthetic heart valve implantation, immunosuppression for cancer or organ transplantation, as well as underlying conditions such as diabetes mellitus, regard a large percentage of individuals today.

The retrospective studies of Arbarillo et al., Nwanyanwu et al., and Yelin et al. [60,61,62] suggested that the true prevalence of infections caused by lactobacilli is higher than estimated only on the basis of case reports. In addition, in many of the reports analyzed here it was stated that lactobacilli are often disregarded as infective agents since they are considered culture contaminants [24,34,61].

Another obstacle for the correct estimate of lactobacilli infection prevalence is the missing reference to identification methods in some reports that were mainly focused on symptom description and treatment for which partial routinary identification and antibiotic sensitivity testing was considered sufficient. As a result, many of the reports stopped identification at the “*Lactobacillus* spp.” level. Consequently, it is not possible to attribute with certainty those cases to lactobacilli also because a strictly related bacterium, *Eggerthia catenaformis*, considered to belong to the old genus *Lactobacillus* until 2011, is capable of causing infections [76,77,78,79] and can be confused with lactobacilli at the phenotypic level [80]. The studies that reported identification to the species level without reference to the identification method (Table 1) probably used standardized colorimetric tests or MALDI-TOF mass spectrometry, largely applied in clinical laboratories [81].

To date lactobacilli involved in infections have been little characterized at genome level but from the studies reviewed here it emerged that the ability to behave as a pathogen is inherent to the strain or clone, besides depending on patient’s underlying conditions and risk factors. Studies on isolate characterization indicated the capacity to form biofilm as the most relevant virulence factor [30,62]. Strikingly, this trait varied between clones of the same probiotic *L. rhamnosus* GG, indicating that the genetic stability of probiotic strains must be checked to avoid use of variants with newly acquired hazardous traits. Though it was demonstrated that mutations can arise in vivo, a higher number of mutations were already present in the probiotic administered to patients [45,62]. Therefore, genome re-sequencing should be applied frequently to probiotic strains at the stage of production to exclude the distribution of genetic variants with increased virulence. This goes further than the current requirements of regulatory bodies that demand genome sequencing for in silico analysis of inherent risk characters of probiotics before further evaluations for use in food, food supplements [82], or LBPs [83]. Indeed, according to the European Pharmacopoeia and FDA guidelines, what is currently considered a satisfactory demonstration of safety for LBPs, that should be extended to all probiotics used in food and food supplements, includes correct strain identification and characterization by phenotypic and genotypic tests, documentation on the origin of the strain (e.g., most recently, the health status of the original donor based on comparative metagenomics), details on passages and manufacturing, analytical methods used to ensure identity, purity and efficacy, history of use, data on the presence of toxins, virulence factors, AMR, production of metabolites that can interact with drugs and of biogenic amines, and studies on toxicity and translocation ability in animal models before clinical trials. Moreover, a risk analysis is carried out considering risks intrinsic to the strain based on whole-genome sequence and information from the literature on potential risk factors for the intended population [84]. Periodical genome re-sequencing for probiotics virulence re-assessment and analysis of genetic stability is not yet a requirement. However, it could allow to exclude genetic mutations or rearrangements that influence virulence and to use only safe variants of the probiotics. This could be more beneficial for health than just avoiding use of probiotics in presence of risk factors.

Given the apparent increase of lactobacilli infection frequency shown in this survey, it would be beneficial to establish an active vigilance to identify cases attributable to these bacteria in healthcare settings, isolate strains, and characterize them for the presence of virulence traits. This could lead to a better selection of probiotic lactobacilli or their variants that can be safely administered also to vulnerable subjects.

Particular attention should be paid to traditional fermented products as possible sources of lactobacilli able to cause infections. The development of autochthonous cultures with well-defined characteristics and their use to prevent the predominance of adventitious strains with unknown characteristics would improve safety of these foods.

## 7. Conclusions

This case report survey indicated an increase of lactobacilli infections and this trend might continue in the future for a more frequent lactobacilli probiotic consumption and increased incidence of diabetes and number of immunocompromised patients, if the situations remains rather uncontrolled as it currently is. Diagnosis of lactobacilli infections allows to adopt appropriate antibiotic treatments that are able to prevent life-threatening evolution of the disease.

Analytical tools for the correct identification of the species should always be applied to the etiological agents in order to define the true prevalence, altered appearance in different categories of patients, and a correct estimate of the risk of lactobacilli infections.

Genetic characterization of strains that caused infections by whole genome sequencing should be undertaken to identify virulence traits and select novel probiotics, or variants of known probiotics, devoid of those traits.

## Figures and Tables

**Table 1 nutrients-14-01178-t001:** Number of case reports per type of infection caused by lactobacilli each year since 2019, with age range of patients, underlying conditions, and species identity of the etiological agents. The number of cases per risk factors and underlying condition are reported separately, though in some instances different risk factors and underlying conditions co-occurred in the same patient.

Year	2019	2020	2021
Endocarditis			
n. cases	6 [14,15,16,17,18,19]	10 [20,21,22,23,24,25,26,27,28,29]	7 [30,31,32,33,34,35]
range of patients age	39–75	40–83	48–83
causative agents	*Lacticaseibacillus paracasei*, unidentified lactobacilli	*Lactobacillus acidophilus*, *L. jensenii*, *L. paracasei*, *L. rhamnosus*, unidentified lactobacilli	*L. jensenii*, *L. paracasei*, *L. rhamnosus*
Risk factors	6 cases: prosthetic aortic valve 1 case: dental extraction 1 case: septic shock due to acute cholecystitis 1 case: intravenous drug abuse	3 cases: dental problems (tooth extraction, teeth scaling, caries) 4 cases: diabetes mellitus 1 case: suspected undiagnosed structural heart disease	3 cases: none 1 case: aortic stent placement 1 case: transcatheter aortic valve implantation 1 case: mitral valve repair
Underlying conditions	2 cases: Birt–Hogg–Dube syndrome 3 cases: none	1 case: pancytopenia, cirrhosis, Crohn’s disease 1 case: Erdheim–Chester disease on chemotherapy 1 case: gastroesophageal reflux	1 case: cardiac disease 1 case: hypertension, obstructive sleep apnea
Bacteremia ^1^			
n. cases	7 [36,37,38,39,40]	1 [41]	4 [42,43,44,45]
range of patients age	neonate-62	75	Neonate-72
causative agents	*L. acidophilus*, *L. rhamnosus*, *Limosilactobacillus reuteri*, unidentified lactobacilli	*L. rhamnosus*	*L. rhamnosus, Lactiplantibacillus plantarum*
Risk factors	1 case: urinary tract infection (UTI), 1 case: treatment with Nivolumab	dental scaling, immunosuppression for renal transplantation	1 case: pre-term birth, CVC 2 cases: treatment with probiotics 1 case: consumption of fermented vegetables 1 case: consumption of home-made yogurt
Underlying conditions	2 cases: diabetes mellitus 1 case: lung cancer	diabetes mellitus	1 case: aortic coarctation 1 case: mild hypertension and colon adenocarcinoma 1 case: HIV infection, Crohn’s disease

^1^ Bacteremia cases considered here are those designated primarily as such by the authors of the case report.

**Table 2 nutrients-14-01178-t002:** Infections different from endocarditis and bacteremia caused by lactobacilli in years 2019, 2020, and 2021.

Type of Infection	Age, Sex	Risk Factors	Underlying Conditions	Causative Agent
2019				
Meningo-encephalitis and bacteremia [46]	63 male			*L. plantarum*
Septic shock [47]	54 male	Consumption of probiotic yogurt	promyelocytic leukemia in second complete remission	*L. rhamnosus* GG
UTI [48]	49 male			*L. delbrueckii* subsp. *delbrueckii*
Perinephric abscess [49]	52 male		diabetes mellitus, obesity, mild hydronephrosis	Unidentified lactobacilli *
2020				
Interstitial pneumonia [50]	68 female	*L. paraplantarum* probiotic supplementation	pancreatic cancer, diabetes mellitus	No cultures carried out
Lung abscess [51]	14 male	Possible aspiration of lactobacilli from yogurt	cerebral palsy, epilepsy and asthma treated with corticosteroids	*L. rhamnosus*
Renal and perinephric abscesses [52]	26 female	Interventions to treat nephrolithiasis		*L. jensenii*
Prosthetic joint infection [53]	82 female	Hip arthroplasty	Past nephrectomy, asthma, hypertension, dyslipidemia, hypothyroidism	*L. paracasei*
2021				
Masticator abscess [54]	23 female	Wisdom tooth extraction	diabetes mellitus	Unidentified lactobacilli
Prostatic abscess [55]	57 male		diabetes mellitus, hypertension	*L. jensenii*
Liver abscesses [56]	59 male	Multiple abdominal surgeries with modified biodigestive anatomy	diabetes mellitus	*L. gasseri*
Pancreatic necrosis and retroperitoneal abscess [57]	88 female		diabetes mellitus, hypertension	*L. paracasei*
Cavernosal abscess [58]	63 male		diabetes mellitus	*L. paragasseri*

* bacterial isolates designated as “*Lactobacillus* spp.” but not identified to the species level.

## Data Availability

Not applicable.

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
