# Peer review of "Lactobacilli Infection Case Reports in the Last Three Years and Safety Implications"

_nutrients, 2022, doi:10.3390/nu14061178_

Round 1

Reviewer 1 Report

The manuscript is devoted to lactobacilli infections case reports and it is well written, illustrated by significant literary materials. It corresponds to the profile of the journal.

In line 15, erase the dot before the word including.

Author Response

We are very very grateful to the reviewer for considering our work of good quality. Done as suggested for Line 15.

Reviewer 2 Report

The topic is very crucial and the paper is well written, but there are some points that need to be taken into consideration:

  • Diagnosis of lactobacillary diseases need to be more considered and appropriate treatments are required to be introduced in life-threatening cases.

  • Its clinical significance need to be more specifically defined. How the clinical appearance of lactobacillus infections is altered?

  • Some certain circumstances such as immunosuppression need to be more discussed. What kind of other susceptible patients should be advised against excessive consumption?

  • Is there any suggested screening protocols for safety of lactobacilli?

  • Antimicrobial Resistance and gene Transferability are also need to be more considered.

  • What is expected for the future regarding the increase in infections caused by Lactobacilli?

  • What kind of medical managements and categorized strategies are required for treatment of unusual cases?

  • At the end of the paper, it would be better to put a separate section entitled “General comments and concluding remarks“.

Author Response

We greatly appreciate the useful suggestions of the reviewer on better underlining some aspects of the subject and indications for improvement of the manuscript. We dealt with the suggestions as described below. The points raised by the reviewer are labelled with “C” and our responses with “R”.

  1. C) The topic is very crucial and the paper is well written, but there are some points that need to be taken into consideration:

Diagnosis of lactobacillary diseases need to be more considered and appropriate treatments are required to be introduced in life-threatening cases.

  1. R) thanks to the reviewer for stressing this point. The above concept was introduced in the newly created Conclusions section (Lines 452-454).
  2. C) Its clinical significance need to be more specifically defined. How the clinical appearance of lactobacillus infections is altered?
  3. R) we agree with the reviewer on the fact that clinical significance of lactobacilli infections needs to be better defined. We underlined in the original version that vigilance plans should be implemented for a better understanding of the true prevalence of lactobacilli infections and also altered appearance in different categories of patients (Lines 439-442; 455-457).
  4. C) Some certain circumstances such as immunosuppression need to be more discussed. What kind of other susceptible patients should be advised against excessive consumption?
  5. R) we discussed this aspect, indicating possible susceptible patients with immunosuppression (Lines 341-349) and pertinent references (70, 71, 72).
  6. C) Is there any suggested screening protocols for safety of lactobacilli?
  7. R) we mention the safety screening process currently in use at Lines 422-434 and added an appropriate reference (84).
  8. C) Antimicrobial Resistance and gene Transferability are also need to be more considered.
  9. R) We underlined that this problem did not seem to occur in the case reports summarized in our review. The reasons are explained at Lines 367-376 and references 73, 74 and 75 were added.
  10. C) What is expected for the future regarding the increase in infections caused by Lactobacilli?
  11. R) An increase of lactobacilli infections is possible for an increase of probiotic consumption and increase in diabetes and immunosuppression incidence, if the situations remains uncontrolled as it is now (Lines 449-452).
  12. C) What kind of medical managements and categorized strategies are required for treatment of unusual cases?
  13. R) We believe that increased awareness on the possibility of lactobacilli infections could improve patient treatment outcome. From the cases described it appears that medical management must rely on prompt diagnosis and treatment with the correct antibiotics to mitigate or prevent severe disease (Lines 362-366).
  14. C) At the end of the paper, it would be better to put a separate section entitled “General comments and concluding remarks“.
  15. R) done accordingly; a “Conclusions” (named accordingly to the journal’s rules) section was added.